# Tract-specific statistics based on diffusion-weighted probabilistic tractography

Andrew T. Reid [1✉], Julia A. Camilleri[2,3], Felix Hoffstaedter [2,3] & Simon B. Eickhoff[2,3]

Diffusion-weighted neuroimaging approaches provide rich evidence for estimating the structural integrity of white matter in vivo, but typically do not assess white matter integrity for connections between two specific regions of the brain. Here, we present a method for deriving tract-specific diffusion statistics, based upon predefined regions of interest. Our approach derives a population distribution using probabilistic tractography, based on the Nathan Kline Institute (NKI) Enhanced Rockland sample. We determine the most likely geometry of a path between two regions and express this as a spatial distribution. We then estimate the average orientation of streamlines traversing this path, at discrete distances along its trajectory, and the fraction of diffusion directed along this orientation for each participant. The resulting participant-wise metrics (tract-specific anisotropy; TSA) can then be used for statistical analysis on any comparable population. Based on this method, we report both negative and positive associations between age and TSA for two networks derived from published meta-analytic studies (the "default mode" and "what-where" networks), along with more moderate sex differences and age-by-sex interactions. The proposed method can be applied to any arbitrary set of brain regions, to estimate both the spatial trajectory and DWI-based anisotropy specific to those regions.

[1] School of Psychology, University of Nottingham, Nottingham, United Kingdom. [2] Institute for Neuroscience and Medicine (INM-7), Jülich Research Center, Jülich, Germany. [3] Institute of Systems Neuroscience, Medical Faculty, Heinrich Heine University, Düsseldorf, Germany. ✉email: andrew.reid@nottingham.ac.uk

Diffusion-weighted imaging (DWI) is a promising non-invasive in vivo technique for evaluating the integrity of myelinated axonal projections in the brain. DWI is based on the attenuation of the T2-weighted MRI signal in the presence of a field gradient, which indicates the degree to which diffusion is unrestricted in brain tissue in the direction of that gradient. Methods that reconstruct DWI maps across multiple gradient orientations can be used to model the (apparent) diffusion of water molecules in discrete compartments (or voxels) of brain tissue, and the anisotropy of this diffusion can be used to estimate the orientation(s) along which diffusion is biased in that voxel. Because myelin is highly lipid-based, and forms a strong hydrophobic barrier, the degree of directed anisotropy in a white matter voxel is presumed to indicate the degree to which it is comprised of coherently oriented myelinated axons[1]. Moreover, variability in this anisotropy can be used to estimate the relative integrity of myelinated fibres that run through a voxel – with the assumption being that decreased anisotropy indicates a decrease in myelination or myelinated axons. Supporting this, evidence from studies of induced degeneration in rat and frog axons resulted in substantial (approximately two-fold) decreases in anisotropy[2,3].

The simplest model of diffusion used for DWI analysis is the diffusion tensor, which assumes a Gaussian distribution with one principal and two secondary axes, corresponding to the first three eigenvectors of observed diffusion across gradient orientations[4,5]. From this model, we can obtain a summary measure of anisotropy (fractional anisotropy; FA), based on the relative magnitudes of the eigenvalues associated with each axis of the tensor. FA ranges from 0 to 1, where 0 indicates perfect isotropy (as would be expected of a uniform substance such as water or cerebrospinal fluid), and 1 indicates diffusion exclusively in the principal direction. The diffusion tensor can also be used to perform deterministic tractography, in which streamlines are generated by starting at a pre-specified set of "seed" voxels, and propagated through a series of neighbouring voxels by reorienting at each according to its principal orientation of diffusion. Tractography can also be set in a probabilistic framework, by generating many streamlines and sampling at each voxel from a posterior probability distribution of orientations based on the tensor model[6].

While the basic diffusion tensor is an adequate model for voxels through which fibres are essentially oriented in a single direction (e.g., fibres traversing the corpus callosum), it fails to model more complex situations, such as the case where two or more fibres are crossing, diverging, or converging. This presents a strong bias in favour of finding certain pathways over others. One way of addressing this issue is to explicitly model multiple fibre directions, for example by using Bayesian model estimation to determine whether multiple fibre orientations are present, and if so how strongly they contribute to the observed diffusion signal[7]. Such an approach greatly improves the ability of probabilistic tractography to discover tracts which traverse areas of uncertainty. As an example from Behrens et al.[7]: while seeding in the internal capsule led only to a prominent primary motor projection in the single-fibre approach, numerous other cortical targets were reached when crossing fibres were explicitly modelled.

It is often desirable to relate voxel-wise DWI-based metrics such as FA to other phenotypical observations, such as behavioural or cognitive measures, or clinical status. Voxel-wise analyses can be highly confounded by the individual geometry of white matter tracts, and one way to address this issue is tract-based spatial statistics (TBSS), in which FA measures are projected onto a population-based FA "skeleton" with a high probability of being white matter in all participants[8,9]. The presence of crossing fibres, however, also has implications for the interpretation of FA[10]. As a hypothetical example: for two otherwise anatomically identical white matter fibres, the introduction of perpendicularly oriented axons to one will reduce its FA proportionally. Interpreting FA in terms of the underlying microstructure of white matter in a voxel is thus inherently ambiguous. This ambiguity can be improved if crossing fibres are explicitly modelled, for example using the Bayesian approach described above. Such a crossing fibre model has been proposed as an extension to the TBSS approach[10].

Additional fibre-specific DWI approaches have also been proposed, including q-space and q-ball imaging, spherical deconvolution, and CHARMED[11–13]. In particular, spherical deconvolution uses a spherical harmonic decomposition to estimate an orientation distribution function (ODF) from the observed diffusion signal[12]. This approach allows both the diffusion model and the number of distinct fibre populations within a voxel to be estimated from the observed data, and is the basis for voxel-wise estimation of apparent fibre density (AFD) for these distinct populations[14]. Differences in oriented clusters of AFD (also referred to as *fixels*;[15]) have been shown for patients with motor neurone disease[14] and Alzheimer's disease[16]. A related spherical deconvolution-based approach, called hindrance modulated orientational anisotropy (HMOA), uses the amplitude of specific lobes of the ODF as an estimate of white matter integrity for a specific fibre population[17]. HMOA of the postcommissural fornix has been shown to predict verbal memory performance in a healthy aging cohort[18].

Although TBSS and fixel-based approaches provide a means of assessing the spatial distribution of statistical effects on white matter integrity, it is often difficult to apply this distribution to specific axonal connections (i.e., between two arbitrary regions of grey matter) in the brain. Suppose, for instance, that we are interested in whether the white matter comprising the physical connection between brain regions $R_a$ and $R_b$ is altered in condition $C$. Using one the aforementioned approaches, we observe that the $C+$ group has decreased FA/AFD/HMOA in an area of white matter that *could* be intermediary to $R_a$ and $R_b$. However, as most major white matter tracts host a mixture of numerous projection, association, and commissural fibres, we can only really speculate about the possibility of a compromised $R_aR_b$ tract. To improve interpretability, we require an explicit approximation of the geometry of tract $R_aR_b$, and an estimate of the diffusion specifically oriented along this tract.

In this study, we introduce a novel methodology to address both of these issues. We first perform probabilistic tractography on a representative sample of participants ($N = 130$, aged 18–80), using high angular resolution DWI data from the Nathan Klein Institute Enhanced Rockland sample[19], and two sets of regions-of-interest (ROIs) obtained from published meta-analytic neuroimaging studies. For each pair of ROIs $R_a$ and $R_b$, we then compute the probability of a tract passing through a given voxel, and use a heuristic approach to determine whether a tract likely exists between $R_a$ and $R_b$, and what its most probable trajectory is. Finally, we use an approach similar to Behrens et al.[7] to determine for each participant, and each voxel, the degree of diffusion in the direction of the tract at that voxel. This yields a tract-specific anisotropy (TSA) metric that can be regressed against variables of interest. Here, we report the tract-wise and 3D distributions of TSA regressed against age, sex, and their interaction.

## Results and discussion

**Tract determination**. We used a heuristic approach (see Fig. 1) to determine the most probable trajectory of a white matter projection between two regions of interest (ROIs). For a given pair of ROIs $R_a$ and $R_b$, we performed probabilistic tractography twice, seeding in one of these regions and terminating in the other. The

## Probabilistic Tractography

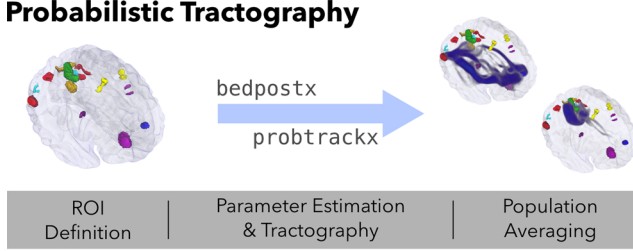

| ROI Definition | Parameter Estimation & Tractography | Population Averaging |

## Tract Trajectory Estimation

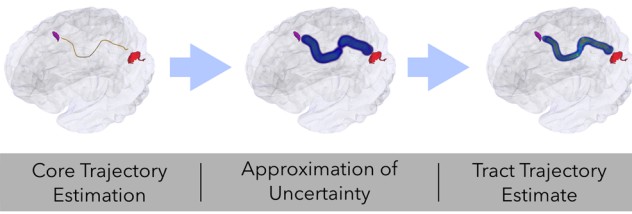

| Core Trajectory Estimation | Approximation of Uncertainty | Tract Trajectory Estimate |

## Tract-Specific Anisotropy

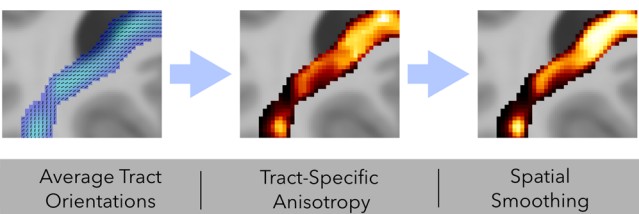

| Average Tract Orientations | Tract-Specific Anisotropy | Spatial Smoothing |

**Fig. 1 Schematic of the procedure.** *Probabilistic tractography*: ROIs were obtained from previous meta-analytic studies and used as seed/target regions for diffusion tensor modelling with `bedpostx` and probabilistic tract tracing performed with `probtrackx`, across all participants. The resulting probability distributions were averaged across directions for each ROI pair ($P_{ab}$). *Tract trajectory estimation*: For each ROI pair, a "core" trajectory was estimated from these bidirectional averages, and represented as a 3-dimensional polyline. An uncertainty field ($\Phi_{ab}$) was then generated from this polyline using an anisotropic Gaussian kernel. Finally, a "core" tract estimate ($P_{ab-\text{tract}}$) was generated as the element-wise product of $P_{ab}$ and $\Phi_{ab}$. *Tract-specific anisotropy*: Average tract orientations were computed for each voxel in a given tract, and these orientations were then regressed against the diffusion evidence for each individual participant. This produced a tract-specific anisotropy (TSA) distribution for each participant, that can be regressed against variables of interest.

resulting directed probability distributions were averaged to obtain a bidirectional probability average, which was then thresholded in order to identify a set of contiguous voxels connecting the two regions. If this step did not result in such a pathway, a connection between $R_a$ and $R_b$ was rejected; otherwise, we identified the "core" of the pathway and used this to obtain a final bidirectional tract trajectory estimate. Tract determination was carried out on two sets of ROIs, derived from previously published meta-analytic studies: the default mode network (DMN), and the what-where network (WWN).

Figure 2a illustrates the tract trajectory estimation process for example tracts PFCm(R) - LOC(R) and dPMC(L)-vPMC(R). The horizontal and coronal slice renderings show the initial minimal bidirectional averages (top row) and final tract trajectory estimates (bottom row). Figure 2b shows three-dimensional renderings of these steps, including the failure to estimate a core trajectory for SPL(L)-dPMC(R). For the vast majority of tracts, this method was able to isolate a single, core trajectory from a variety of alternatives.

For the DMN, 33 of 36 ROI pairs (92%) produced core tract estimates (Fig. 3). Two of the failed tracts involved the left PFCm, with the more posterior LOC(R) and PCm(L) regions. Tracts connecting PCm and PCG to PFC and ACC traversed the more superior cingulum bundle, while those connecting LOC to PFC and ACC traversed the more inferior fronto-occipital fasciculus. Contralateral DMN connections traversed the splenium of the corpus callosum (CC; Fig. 3d) for posterior ROIs, and the genu of the CC for anterior ROIs and anteriorly projecting LOC connections.

For the WWN, 43 of 45 tracts were estimated (96%), with only 2 tracts failing because no core tract could be identified (Fig. 3). The failed tracts were the two contralateral connections between SPL and dPMC (Fig. 3c). Successfully estimated tracts consisted of anteroposterior projections traversing the superior longitudinal fasciculus, and contralateral connections traversing the splenium of the CC and the inferior part of the body of the CC (Fig. 3d).

ROI pairs failed to generate tract trajectory estimates when the thresholding applied to the bidirectional population average distribution broke the contiguous path between them. Notably, this is not evidence that the tract does not exist, but rather that there is ambiguity about the location of its trajectory, based on the diffusion evidence. It is important to note that the threshold applied here was determined somewhat arbitrarily, i.e., by observing what value reduced the number of alternative pathways to a single, core one. In cases where this failed, there were typically two or more alternatives that could not be disambiguated (see Fig. 2b for an illustration of failed tract SPL(L)-dPMC(R)). Further investigation into these ROI pairs, e.g., by comparing them to known connectivity evidence from tract tracing studies (in non-human primates) or different modalities (in humans), may be useful for determining whether a tract indeed exists between them.

Previous studies have also evaluated white matter tract geometry using DWI. In a now-classic study, Catani et al.[20] used deterministic tractography to identify and "dissect" individual tracts based on seed ROIs. This was extended by Jones et al.[21] in order to map diffusion metrics onto the trajectories of specific tracts. More recently, Colby et al.[22] used B-spline resampling to map FA to specific tracts in individual participants, allowing effects to be mapped to points along its trajectory. The present approach extends the work of these previous studies, in that it facilitates the investigation of white matter pathways connecting specific ROI pairs, rather than coarse-scale fasciculi, in terms of tract geometry and anisotropy estimation. Another important advantage is its use of a probabilistic tractography framework to describe the "core" trajectories of specific tracts across a population of interest.

**Tract-specific anisotropy**. Given a tract trajectory estimate, as described above, we next wanted to determine how strongly diffusion profiles of individual participants were oriented along that tract. For a given voxel, we determined the average streamline orientation derived from the previous probabilistic tractography step, and then computed, for each participant, a "tract-specific anisotropy" (TSA) estimate, indicating the degree to which this average orientation loaded onto the DWI intensities along each gradient direction (TSA values are the $\beta$ parameters estimated from Eq. (3)).

Kernel density estimates for TSA, for both networks, are shown in Fig. 4a. These show the distribution of scores for tracts thresholded at $P_{ab-\text{tract}} > 0.5$. While most distributions were roughly Gaussian, with a heavy positive tail, this varied across tracts in terms of both kurtosis and skewness, and some shorter tracts (e.g., PCG(L)-PCm(R)) had very little variance. For a

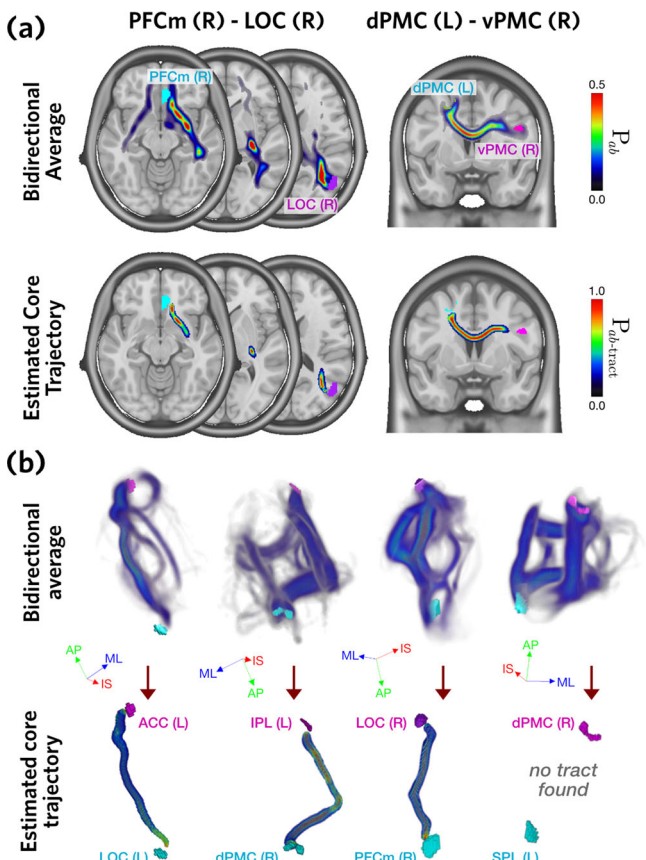

**Fig. 2 Tract trajectory estimation. a** Core tract trajectory estimation steps, shown for two exemplar tracts. The top row shows the streamline probability determined across all participants ($n = 130$), averaged in both directions, $P_{ab}$. The bottom row shows the estimated core trajectory for each ROI pair, $P_{ab-tract}$. **b** Three-dimensional distributions of $P_{ab}$ (top) and $P_{ab-tract}$ (bottom), for four exemplar tracts. Axis labels: ML medial-lateral, AP anterior-posterior, IS inferior-superior.

number of tracts (e.g., SPL(R)-IPS(L)), a bimodal distribution was evident. The overall distribution for each network is shown in the insets; WWN showed a broader, flattened distribution relative to DMN.

Figure 4b shows the two-dimensional spatial distributions of TSA scores for two exemplar participants, along with the average TSA across all 130 participants. These distributions demonstrate variability between participants, but in general the highest TSA scores were observed in the centre of estimated tract trajectories - reflecting the strongest loading of diffusion profiles - which tapered off towards the edges.

It is common to use streamline counts from probabilistic tractography as an estimate of connection strength between two ROIs (e.g.,[23–25]). This approach, however, suffers from a number of seemingly intractable biases, including: the nature of the diffusion profile through which a given tract traverses (anisotropy bias); the length of the tract (distance bias); and the position of seed and target ROIs relative to gyri or sulci[26,27]. It is thus uncertain how to interpret a relative streamline count, or change in this quantity in association with some covariate of interest, with respect to biophysical properties such as white matter integrity or connectivity strength. The present approach is arguably less susceptible to these biases, because it only models the geometry of streamlines that *do* connect two ROIs. However, future research efforts should be directed at validating (1) the interpretability of TSA with respect to axonal white matter

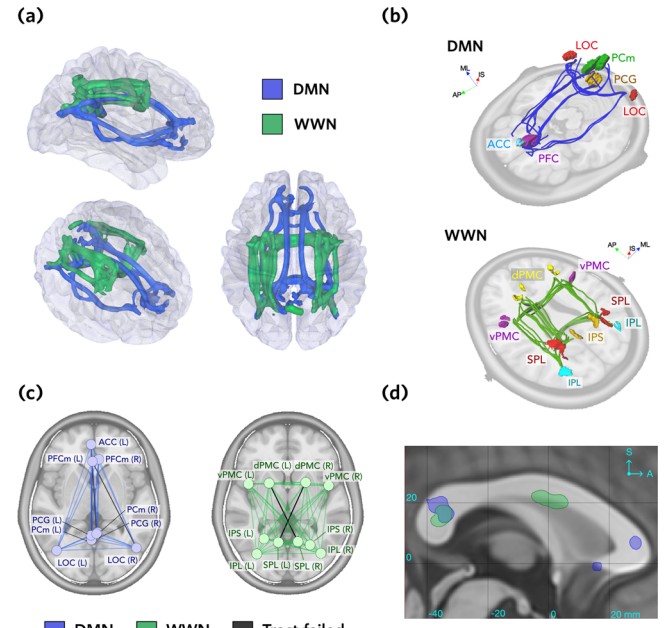

**Fig. 3 Tract trajectories for DMN and WWN. a** All accepted tracts shown for the DMN (blue) and WWN (green) networks, rendered as isosurfaces thresholded at 0.5 (top, right, and oblique perspectives). **b** Core polylines representing the trajectories of accepted tracts in the DMN (top) and WWN (bottom) networks. Images are rendered with oblique perspectives next to planar sections of the ICBM nonlinear T1 template image, for reference. See Materials and Methods for names of the ROIs. **c** Graph representations of the DMN (blue) and WWN (green) networks, with failed edges shown in grey. Graph vertices represent the centre points of each ROI, projected onto the transverse plane. **d** Sagittal section at midline, showing where contralateral tracts (thresholded at 0.5) for each network traverse the corpus callosum. Coordinates are ICBM152. Axis labels: ML medial-lateral, AP anterior-posterior, IS inferior-superior.

integrity, and (2) the spatial specificity of trajectory estimates. This could be done, for example, through the use of phantoms[28–30] or histological approaches[31,32].

Previous studies have investigated DWI metrics in a tract-specific manner. Notably, Hua et al.[33] produced probabilistic maps of 11 gross WM tracts by seeding in WM voxels and averaging deterministic tractography streamlines across participants, generating tract-specific metrics by averaging FA across all voxels in a tract. The well-known TBSS approach[9], in which statistics are projected onto a pre-established population-based white matter "skeleton", similarly allows DWI metrics to be mapped to specific WM tracts. Both of these approaches are similar to the current method in that they utilise population averaged images (streamline counts or FA values) in order to generate probabilistic maps of WM tract geometry (see[34], for a review of such approaches). These maps serve a similar function to population-based anatomical grey matter templates, such as the linear and nonlinear ICBM-152 templates[35,36]. There are two major advantages of the present method over the Hua et al.[33] method and TBSS: (1) it allows a population-based tract estimate to be *derived specifically for the white matter tract connecting any two arbitrarily-defined GM ROIs*, if that tract is likely to exist; and (2) it *allows participant- and tract-specific anisotropy to be estimated*, based on the orientation of streamlines defining that tract in each voxel along its trajectory.

**Tract-specific age and sex effects**. Having obtained TSA scores for each individual participant, we were next interested in

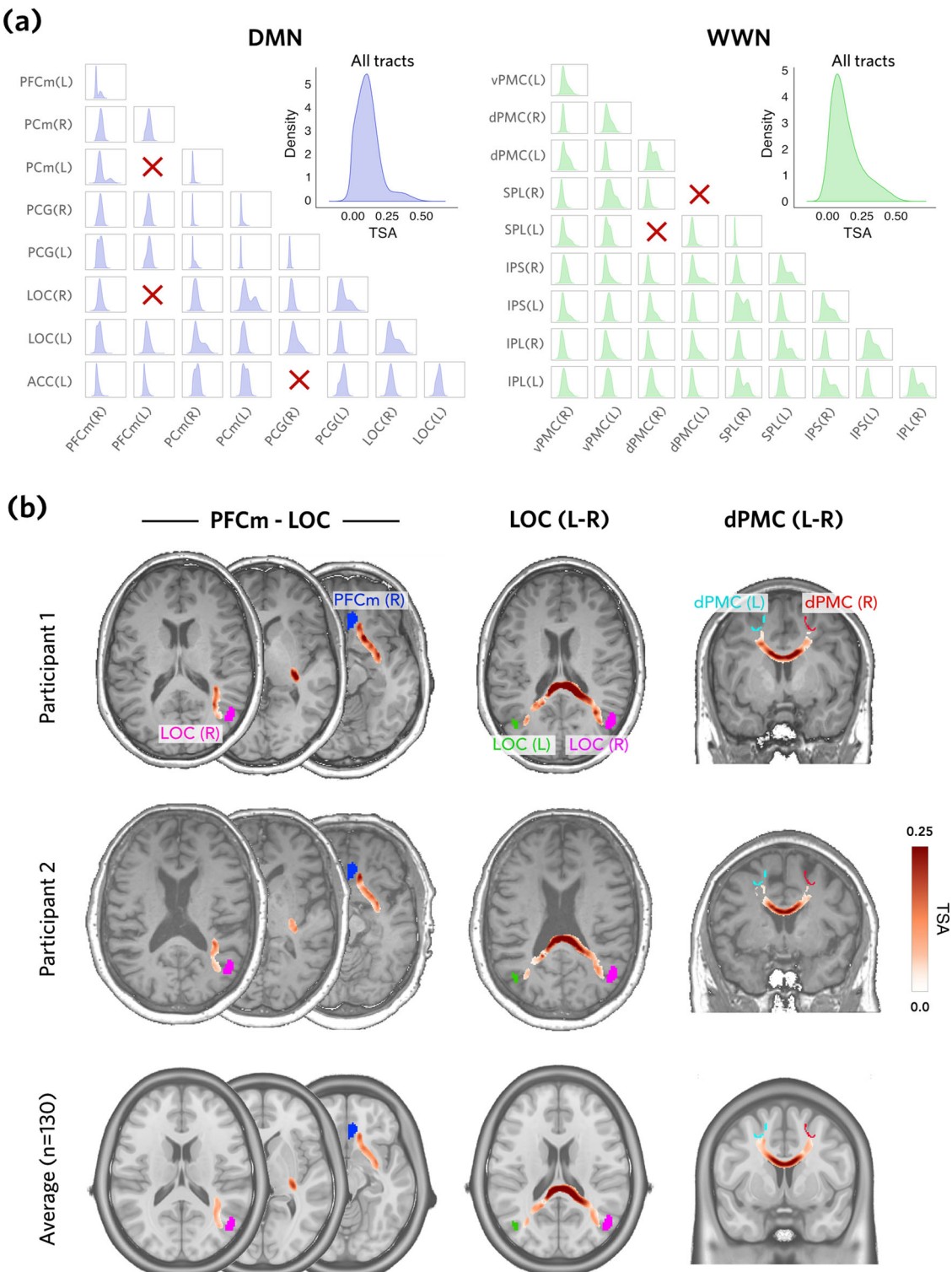

**Fig. 4 Tract-specific anisotropy (TSA). a** Distributions of TSA across voxels and participants, for each estimated tract (ROI pair), shown as kernel density estimates for DMN (blue) and WWN (green). Insets show the distribution of TSA values over all tracts; x-axis scales are the same for all plots. Red crosses indicate that no tract was generated. **b** Spatial distributions of TSA values for two exemplar participants and averaged over all participants, and three examplar tracts, shown in horizontal and coronal section. Anatomical images are the ICBM152 nonlinear template.

whether these scores were associated with age, sex, and their interaction. We performed voxel-wise regressions of the form $TSA = \beta_0 + \beta_1 \cdot Age + \beta_2 \cdot Sex + \beta_3 \cdot Age \times Sex + \varepsilon$. T-statistics for each coefficient were obtained, and summarised at each distance along the tract. For each tract, we then used one-dimensional random field theory to identify significant clusters

of t-statistics ($p < 0.05$). To control for family-wise error, we limited the false discovery rate (FDR) over all tracts to 0.05.

For the DMN, there were significant associations of TSA with age and sex, as well as a (weak, $R^2 = 0.03$) interaction between these two factors (Fig. 5, top row; and Supplementary Table 1). Age effects were fairly diffuse, and ranged in effect size ($R^2$) from

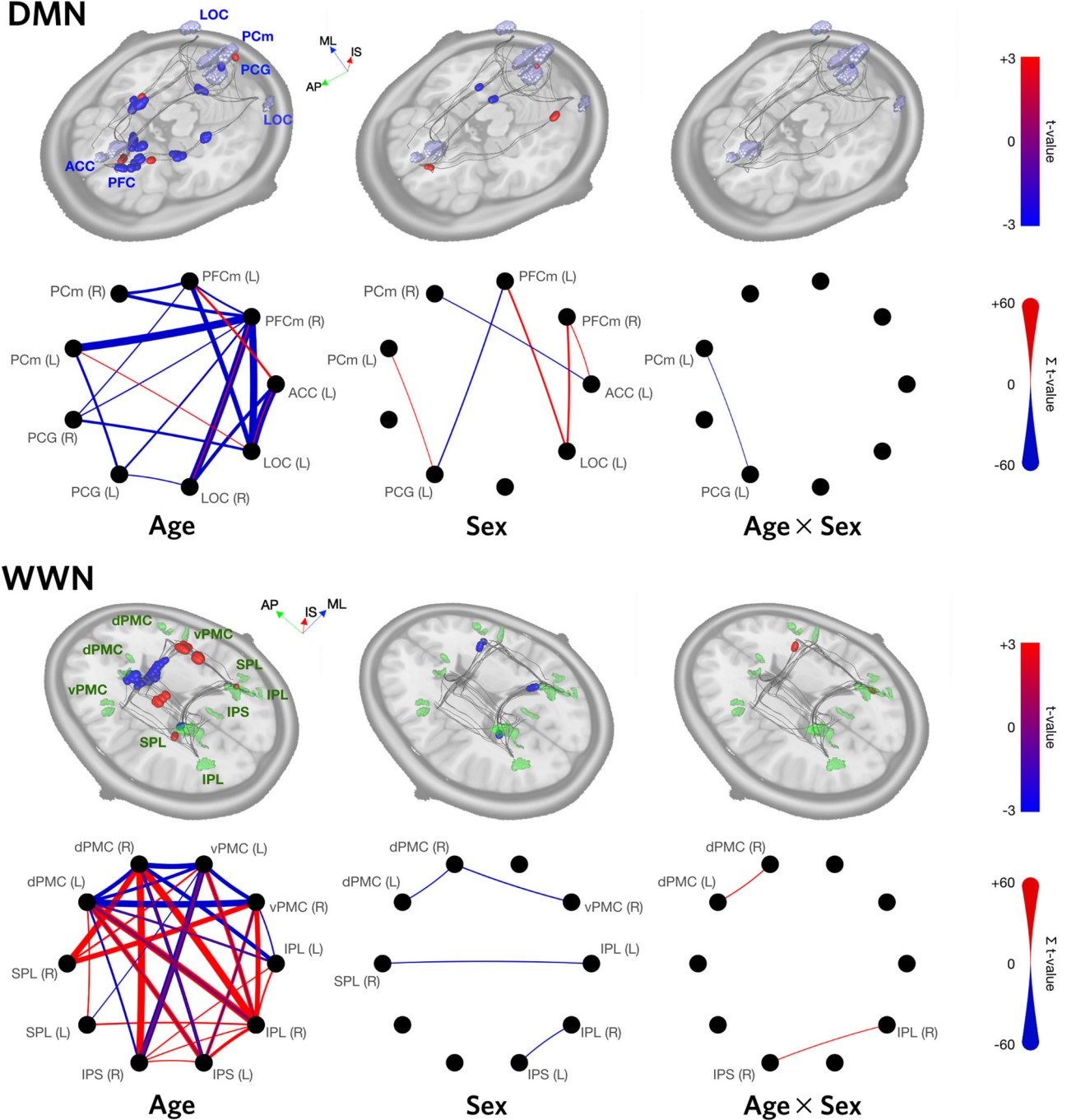

**Fig. 5 Age and sex effects for DMN and WWN.** Regression results are shown for both networks, for linear models of the form $TSA = Age + Sex + Age \times Sex + \epsilon$. 3D renderings show the maximal $t$-values (thresholded using cluster-wise inference, with $p < 0.05$ and FDR $< 0.05$) at each distance along the core trajectories of each tract in the network. Circular graph representations show the sum of significant positive (red) and negative (blue) $t$-values for each tract. The thickness of an edge is proportional to its sum, and the absence of an edge indicates that no significant clusters were found for that tract. In all cases, $N = 130$. Axis labels: ML medial-lateral, AP anterior-posterior, IS inferior-superior.

0.07 to 0.13. Negative effects of age were strongest, and were found in 15/32 (47%) of DMN tracts, including ipsi- and contralateral connections between PFC and LOC (traversing the fronto-occipital fasciculus), and PCm(L)-PFCm(R). Modest positive effects of age were also found for 6/32 (19%) of tracts. Figure 6 shows scatter and violin plots for selected tracts. Notably, tract PCm(L)-PCG(L) showed effects for age, sex, and their interaction; and tract ACC(L)-LOC(L) and PFCm(R)-LOC(R) showed both positive and negative age effects.

The WWN also showed significant associations for age, sex, and their interaction (Fig. 5, bottom row; and Supplementary Table 2). Surprisingly, there were strong positive and negative age effects for this network, with 16/32 (50%) showing negative effects and 14/32 tracts (44%) showing positive effects. Effect sizes for age ranged from 0.05 to 0.11 Compared to DMN, the age effects for WWN were more focal. Negative age effects were found proximal to dPMC(L), and in collosal fibres traversing the body of the corpus callosum (see Fig. 3d), involving contralateral

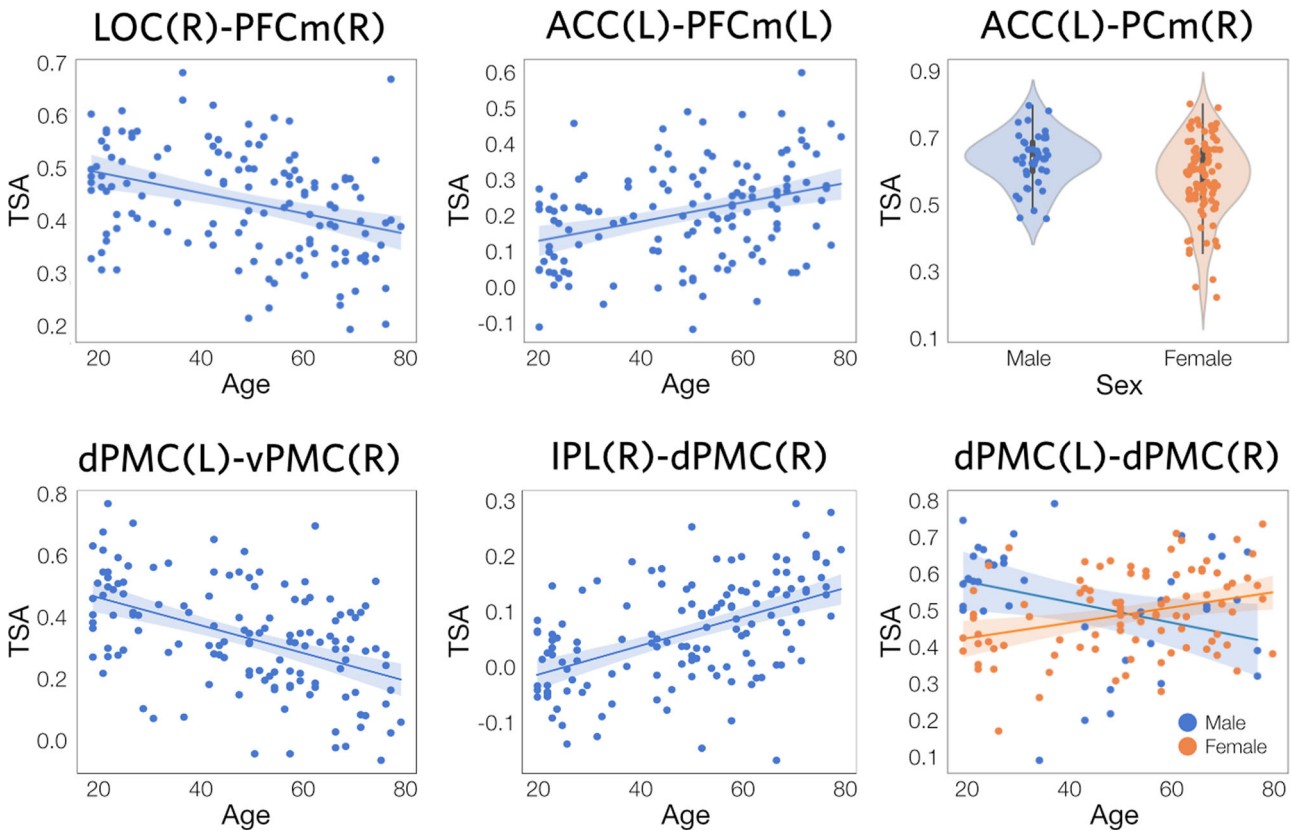

**Fig. 6 Scatter and violin plots of age and sex effects.** Plots of selected TSA regression results are shown ($N = 130$). *Top row*: DMN tracts with (from left to right) negative and positive age effects, and a sex difference (male > female). *Bottom row*: WWN tracts with negative and positive age effects, and an Age × Sex interaction. Data points are the mean TSA values within significant clusters (as shown in Fig. 5). Shaded areas on scatterplots represent 95% confidence intervals.

tracts between dPMC and vPMC. There were also negative associations found in contralateral tracts between IPS, dPMC, and vPMC. The largest positive age effects occurred along anteroposterior-oriented tracts in the longitudinal fasciculus, particularly those involving left and right dPMC. All sex differences were negative (male > female) for this network, on were found on contralateral tracts. Figure 6 shows scatter and violin plots for selected tracts. Notably, the homotopic dPMC(L)-dPMC(R) tract showed both a sex difference and an Age × Sex interaction.

Scatter and violin plots for all significant effects, and line plots and tables showing raw and cluster-thresholded *t*-values for all tracts, are provided in Supplementary Figs. 3, 4, and 5.

There is an abundance of evidence for age-related decreases in DMN connectivity, including a decrease in fMRI-based resting-state functional covariance[37–39], and age-related decreases in FA in white matter tracts proximal to DMN regions[38]. Marstaller and colleagues found a reduction in the extent to which posterior cingulate and precuneus activity covaried with wider brain networks, along with reduced FA in numerous tracts, including the fronto-occipital fasciculus, where the current negative age association is most prominent. Sambataro et al. reported decreased covariance between PCC and PFC, which predicted working memory performance. Decreases in DMN functional covariance also appear to be accelerated in Alzheimer's disease (also reviewed in[40,41]).

For the WWN, age-related decreases in TSA occurred mainly in the body, but not the splenium, of the corpus callosum. This pattern is in agreement with several DWI-based studies of age-related connectivity changes. Burzynska et al.[42] used TBSS to show an age-related reduction in FA (and increase in radial and mean diffusivity) in the genu and body of the corpus callosum, but not the splenium. Using a DTI approach, Bennett et al.[43] found decreased FA in older versus younger participants for both the genu and splenium, but with a more pronounced effect in the former. The same authors report an anterior-to-posterior gradient in age-related FA changes, with these being more pronounced in frontal white matter, consistent with the pattern found in the current study[44]. An even more pronounced pattern was reported by Michielse et al.[45], who found an age-related decrease of FA in the genu, no relationship in the body, and a late (age 70–85) *increase* in the splenium. Bastin et al.[46], also using tract shape modelling, found a significant decrease in FA for the genu, but not the splenium, in an elderly cohort (age 65–87), while in a cohort ranging from 30 to 80 years, Hsu and colleagues[47] reported a similar age-related decrease in FA (and increase in MD) for the anterior, but not posterior corpus callosum. Interestingly, evidence from Hasan et al.[48] suggests FA in the corpus callosum changes in a *quadratic* manner across the lifespan: increasing between age 7 and 20 years and decreasing between 20 and 60, which is in line with the present findings. Taken together, these findings suggest that, in adults, age-related changes in WM integrity of the corpus callosum may be more prominent anteriorly, and reduced or even reversed posteriorly.

Positive age associations were a more surprising finding, as numerous articles report negative age/FA associations (e.g.,[49]) and postmortem evidence of white matter loss and decrease in the proportion of small myelinated fibres with age[50,51]. Both positive and negative age/FA associations have been reported in at least one previous brain-wide TBSS study[52], however. For DMN in

particular, we found that most positive associations occurred in regions proximal to ROIs, where the potential confound of crossing fibres is likely more pronounced[53]. The positive relationships in the WWN were especially prominent, and suggest a (paradoxical) increase in white matter integrity in these tracts. The majority of positive relationships in WWN were found in the middle of the superior longitudinal fasciculus (SLF). One TBSS study focusing on the SLF in a healthy cohort found no effect of age on FA[54], while another whole-brain study found SLF among tracts with negative age effects[38]. Similarly, Rojkova et al.[55], using FA and the fibre-specific HMOA approach, found an age-related decreases in HMOA in the SLF (amongst other tracts), and reported that HMOA was more sensitive than FA alone in identifying age-related changes. On the other hand, increased FA in Alzheimer's disease patients has also been reported in SLF[56]. The authors of this study suggest that a relative sparing of crossing motor fibres may account for this effect, but this is inconsistent with our observed increase in TSA; on the contrary, our findings might be explained by a relative *decrease* in WM integrity in these crossing fibres.

It is possible that the increased specificity of the current approach permits a more fine-grained spatial and angular dissection of effects than does TBSS with FA, which uses a more coarse-grain white matter skeleton, and is not orientation-specific. If so, then the positive effects observed here may reflect a real age-related increase in white matter integrity for specific tracts. This possibility is supported by reports of fairly widespread increases in fMRI-based functional covariance with aging[57], which have been proposed to reflect compensatory changes in response to degeneration or dysfunction of other brain regions. Given the conflicting evidence, however – particularly with the HMOA evidence from Rojkova et al.[55], which is fibre-specific – these effects should be interpreted with caution. It will be important in future TSA studies to increase the number of ROIs, or query specific crossing tracts, in order to obtain a more complete picture of age-related effects across white matter. One promising avenue could be based on the so-called "tract-specific fractional anisotropy" approach[30], in which uses the free water fraction to estimate an adjusted FA value in crossing-fibre regions. Alternatively, an integration of the current approach with existing fibre-specific spherical deconvolution-based methods, such as AFD[14] or HMOA[17], could potentially be used to disambiguate the interpretation of TSA values in areas of dense crossing fibres.

More modest sex differences, and age-by-sex interactions, were also observed for both networks. Previous DWI-based studies have also found sex differences in FA, typically with males showing higher FA values[58–60]. For DMN, we found negative effects (males > females) for the PCG(L) with PFCm(L), which accords with findings from Menzler et al.[60], who report prominent differences in the cingulum bundle. We also found positive effects (females > males) that were mostly left lateralized and included LOC(L), which have not commonly been reported in the literature. Although sex differences in LOC function have been hypothesised on the basis of differential object perception[61] – a function that has been associated with this region – there does not currently appear to be much direct evidence for this.

It is also notable that, for numerous tracts, we found both positive and negative age effects in different parts of the same tract (see Fig. 5). It is not immediately clear how to interpret such a result. If our inference is that a TSA value represents the number of intact axons projecting between two grey matter ROIs, this is contradicted by the observation of both increases and decreases in this region - damage to an axon anywhere along its length will trigger Wallerian degeneration in both directions. On the other hand, this finding is consistent with the idea that TSA reflects the degree of (de)myelination, which may be increased on

average in one part of the tract and decreased in another. Importantly, there is evidence that demyelinated axons, while they may be functionally impaired, are not necessarily at higher risk of degeneration[62]. A third possibility is that TSA is influenced by the degree of crossing fibres in a particular region along its length; changes to diffusion in directions other than the average orientation of interest will influence the regression fit used to estimate this metric.

**Limitations and future directions.** The NKI Rockland dataset was chosen due to its large size, age range, and the use of a single MRI scanner and protocol. To ensure the cohort was as representative of the general population as possible, and to enable the analysis of age over the lifespan, we chose to use close to the full age range (18–80), and to exclude participants with clinical diagnoses. As with most population templates, however, the choice of cohort is an important consideration when interpreting a derived result. The human brain is known to show systematic anatomical grey matter changes across the lifespan[63,64], and this will almost certainly bias normalisation in a way that may account for a portion of the TSA effects reported here. Indeed, variability of findings due to the choice of T1w template has been shown for voxel-based morphometry[65]. It will be important in future studies to assess the influence of this bias, use cohorts that are more targeted to a particular phenomenon under investigation (see, e.g.,[66]), and compare the predictions of TSA to in vivo or post mortem analyses of white matter (e.g., as in[26]).

The estimation of tract trajectories and TSA values involves a heuristic approach, with numerous parameters involved at each step. This raises the potential for parameter adjustments to vary the results in ways that bias the resulting distributions and statistics. In general, however, these parameters were chosen in order to spatially constrain these estimates in reasonable ways; for example: to optimise a threshold such that a single trajectory is chosen from several alternatives, to constrain core polylines to realistic geometries, or to determine the spatial extent at which TSA values are used to compute distance-wise statistics. Of these parameters, it is only the latter where the researcher is able to exercise discretion, i.e., over the degree of spatial certainty used to map statistics at a given distance to the "core" tract trajectory (specified by the $\sigma_r$, and $\lambda$ parameters). As a general policy, we recommend using the default parameter values for tract and TSA estimation, in the absence of a principled reason to adjust them.

The relatively small networks used here (9 ROIs for DMN and 10 ROIs for WWN) required a total processing time of over 200 hours per participant, on CPU processors (see Supplementary Table 3). Notably, this processing time will scale quadratically with the number of ROIs ($O(n^2)$), indicating that it may be infeasible to apply the TSA approach to the full set of possible ROIs (of comparable size) in the brain. On the other hand, the volume of grey matter in the brain is finite, and the number of ROIs comprising a whole-brain network depends critically on their *granularity*. Additionally, the bulk of processing time for this approach is attributable to the preprocessing steps (`bedpostx` and `probtrackx`). GPU versions of these functions have recently been introduced, which can reduce processing time by a factor of 200[67]. This would reduce the required processing time for the present study to 10 hours per participant. A related limitation is that the number of multiple hypothesis tests (and associated family-wise error) also increases by $O(n^2)$. However, the false discovery rate (FDR) approach we use to control family-wise error should be robust even to the high number of tests expected with a whole-brain TSA analysis (and is commonly used in genomic studies; see[68]). Ultimately, whether whole-brain TSA analysis is feasible in practice remains to be demonstrated.

The ability to estimate spatial trajectories of white matter projections between specific pairs of ROIs could lead to a number of important applications. For instance, this information could be used to predict the functional outcomes of age-related white matter lesions (WML), which have a prevalence of ~18% in people aged 60–69 and ~40% in people over 80[69]. Conceivably, clinicians would identify WML locations using T2-FLAIR imaging, estimate which tracts intersect these lesions, and predict functional deficits on the basis of the brain regions connected by these tracts. This approach could also augment previous studies investigating connectivity[70], cortical thickness[71], and gait changes related to WML[72].

## Methods

**Participants**. Participant data were obtained from the publicly available Nathan Klein Institute (NKI) Enhanced Rockland sample[19,73], through the 1000 Functional Connectomes Project (www.nitrc.org/projects/fcon_1000/). We included participants from the first four data releases, while excluding any participant with an existing clinical diagnosis at the time of scanning. In total, 130 participants (86 female, age range 18–80) were analysed. Written informed consent was obtained from all participants.

**Neuroimaging data and metadata**. All imaging data in the Rockland sample were acquired from the same scanner (Siemens Magnetom TrioTim, 3.0T). T1-weighted images were obtained using a MPRAGE sequence (TR = 1900 ms; TE = 2.52 ms; voxel size = 1 mm isotropic). DWI was collected with a high spatial and angular resolution (TR = 2400 ms; TE = 85 ms; voxel size = 2 mm isotropic; b = 1500 s/mm$^2$; 137 gradient directions). Age and sex data were also obtained, with the Sex factor encoded as 1 = Male, 2 = Female.

**Preprocessing**. DWI images for all participants were preprocessed using FMRIB software[74,75]; specifically, FSL version 6.0.0. Participant datasets were processed in parallel using a Linux-based SLURM computing cluster, located at the University of Nottingham. Raw diffusion data were first corrected for eddy current artifacts using the `eddy` command. All $B_0$ (zero-gradient) images were then averaged and used to extract a brain mask using the `bet` command. Next, for each voxel in the brain mask, a diffusion tensor model was fit to the data using `dtifit`, and subsequently passed to the `bedpostx` command, which infers the existence of crossing fibres and estimates the contribution of each crossing fibre to the diffusion-weighted signal[7]. The default number of 3 fibre populations was modelled. Finally, to express the voxel-wise diffusion models in standard space, linear and nonlinear transforms between native diffusion and MNI-152 space were estimated, by applying the `flirt` (12 degrees of freedom) and `fnirt` commands, respectively, using individual FA images as input and a mean template FA image as the reference. Inverse transforms were also estimated, using the `invwarp` command, for use in later steps.

**Regions of interest**. Seed regions of interest (ROIs) were obtained from two previously-defined networks (see Fig. 3b). For each network, all possible ROI pairs were considered as potential tracts.

The *default-mode network* (DMN) was derived from Schilbach et al.[76], who performed an Activation Likelihood Estimation (ALE;[77,78]) meta-analysis using 533 experiments from the BrainMap database[79], querying for regions that were consistently deactivated across tasks. The ROIs were obtained by downloading the DMN network from the ANIMA database (https://anima.inm7.de/studies/Schilbach_SocialNetworks_2012;[80]). The network was comprised of 9 ROIs: anterior cingulate cortex (ACC; left only), lateral occipital cortex (LOC; bilateral), posterior cingulate gyrus (PCG; bilateral), precuneus (PCm; bilateral), and medial prefrontal cortex (PFCm; bilateral).

The *what/where* network (WWN) was derived from Rottschy et al.[81], with data obtained from the ANIMA database (https://anima.inm7.de/studies/Rottschy_WorkingMemory_2012). The authors performed a conjunction on ALE analyses for tasks testing memory for object identity ("what"; 42 experiments) and object location ("where"; 13 experiments). The WWN was comprised of 10 ROIs, with 5 brain regions represented bilaterally. These ROIs were: dorsal and ventral premotor cortex (dPMC, vPMC), superior parietal lobule (SPL), inferior parietal lobule (IPL), and infraparietal sulcus (IPS).

**Probabilistic tractography**. Probabilistic tractography was performed in MNI-152 space, using the `probtrackx` command[7]. For both networks, we generated 50,000 streamlines per seed voxel, with the other seed regions as target masks. We applied a step length of 0.5 mm, a curvature threshold of 0.2, a minimal path distance of 5 mm, and a fibre threshold of 0.01. A distance correction was also applied. Notably, this correction merely multiplies the streamline count at each voxel by its expected distance from the seed voxel. This is not an adequate correction for distance bias (which is unlikely to be linear), but does help reduce this bias in the middle of trajectories, for the purpose of tract determination (see next section). For each voxel `probtrackx` recorded the number of streamlines that encountered that voxel, producing a separate count for streamlines terminating in each target ROI. Additionally, the voxel-wise mean orientation of streamlines between each seed/target pair was computed.

**Tract determination**. Tracts were determined separately for each ROI pair. Voxel-wise streamline counts for each direction ($R_a$ to $R_b$ or $R_b$ to $R_a$) were first normalised by dividing by the total number of streamlines reaching target $R_b$ from seed $R_a$, and the minimum across both directions was obtained. These minimum images were subsequently averaged over participants, smoothed, and normalised to the range [0,1], yielding a probability $P_{ab}(i)$ of voxel $i$ being included in tract $T_{ab}$. It is noteworthy that in some cases, the matrix $P_{ab}$ of such values is constrained to a single, geometrically confined tract, while in others, it reflects many possible routes between two ROIs (see examples in Fig. 2b). Our objective was to identify the single most probable tract, or reject the tract altogether if such could not be determined. This was done in a heuristic manner. Firstly, we thresholded $P_{ab}$ by setting all values where $P_{ab} < \alpha$ to zero. A threshold of $\alpha = 0.07$ was applied by experimentation, in order to both disconnect most low-probability alternative pathways, and ensure that (for cases where the existence of a tract is known or likely) at least one pathway remained intact. Additionally, because $P_{ab}$ proximal to ROIs tended to be lower that for the main tract trajectory, ROIs were dilated by 3 voxels to ensure they remained connected after thresholding.

We next discretized $P_{ab}$ into distance assignments D, using a flood-fill approach, assigning all voxel neighbours of seed region $R_a$ a value of $d = 1$, all subsequent neighbours $d = 2$, and so on, until all voxels in $T_{ab}$ were assigned a distance. We then identified the voxel of maximal $P_{ab}$ for each distance $d$, constructing a polyline $L_{ab}$ with centre points of these voxels as its vertices. Inclusion of a vertex in $L_{ab}$ was conditional on two constraints: (1) segment length $|x_{i-1,i}| < 4$ mm; and (2) vertex angle $\theta_i < \pi/3$. Where a constraint was violated, the voxel with the next highest $P_{ab}$ was tested, and so on until a voxel was found satisfying the constraints. In cases where no appropriate voxels were found, the maximal voxel was added and the violation was recorded as a flag for visual inspection. $L_{ab}$ was extended in this fashion until: (1) the target region $R_b$ was encountered, and the tract was accepted for further analysis; or (2) the maximal distance was encountered, but not the target region $R_b$, and the tract was rejected for further analysis (i.e., the thresholding broke all routes, and thus a "true" route between $R_a$ and $R_b$ could not be determined).

With the assumption that an accepted polyline represents the geometric centre of a given tract $T_{ab}$, we then modelled the probability $P_{ab-\text{tract}}$ of a voxel being in that tract, as the product of an uncertainty field $\Phi_{ab}$ oriented around $L_{ab}$, and the original probability field $P_{ab}$ (see Fig. 1).

$\Phi_{ab}$ was constructed as follows. For every vertex $v \in L_{ab}$, an orientation vector $\omega_v$ was computed as the sum of the segment vectors $x_{v-1,v}$ and $x_{v,v+1}$ (at the endpoints of $L_{ab}$, only one segment was used). From $\omega_v$, an anisotropic Gaussian kernel $\phi(i, \omega_v, \mu, \sigma_a, \sigma_r)$ was applied to the subset of voxels within a radius of 8 mm of $v$. Parameters defining the shape of the Gaussian function were fixed at $\mu = 0$ mm, axial $\sigma_a = 10$ mm, and radial $\sigma_r = 4$ mm; where axial and radial axes were parallel and perpendicular to $\omega_v$, respectively. For a given voxel $i$ in $T_{ab}$, the maximal value of $\phi$ across all vertices $v$ was assigned:

$$\Phi_{ab}(i) = \max_v \big( \phi(i, \omega_v, \mu, \sigma_a, \sigma_r) \big) \qquad (1)$$

$P_{ab-\text{tract}}$ was determined as:

$$\mathbf{P}_{ab-\text{tract}} = f(\boldsymbol{\Phi}_{ab} \odot \mathbf{P}_{ab}, d) \qquad (2)$$

The function $f(g, d)$ normalizes $g$, at each discrete distance $d$, to values between 0 and 1.

**Tract-specific anisotropy estimation**. Having defined (or rejected) a tract $T_{ab}$, we were next interested in extracting meaningful diffusion metrics from it, which can allow us to perform statistical inference on specific tracts. At each voxel, we thus wanted to estimate how strongly its diffusion weighed onto the orientation of the tract at that voxel. To do this, we obtained the average orientation (across all participants) of streamlines going through voxels in $T_{ab}$ using `probtrackx` (see Probabilistic tractography). These average orientation images, along with the $P_{ab-\text{tract}}$ images, were warped from standard to individual participants' diffusion space, using the inverse transforms computed in the preprocessing step (see Preprocessing).

Next, for each participant, and for each voxel $j$, the fraction of the diffusion-weighted signal along the average tract orientation was estimated by fitting the following linear regression using the `statsmodels` Python library (https://www.statsmodels.org):

$$\mathbf{s}_j / s_j^0 = \beta_j \cdot e^{-b\delta(\mathbf{R}^\top \overline{\mathbf{v}}_j)^2} + c_j \qquad (3)$$

where $\overline{v}_j$ is the average streamline orientation, R is the $M \times 3$ matrix of gradient orientation vectors (where $M$ is the number of orientations, here $M = 137$), $s_j^0$ is the non-diffusion-weighted signal, $s_j$ is the observed signal at each gradient orientation, $b$ is the gradient strength (b-value), and $\delta$ is the diffusivity.

Notably, this formulation is equivalent to that presented in[6,7], but applying only to the average orientation $\bar{v}_j$. $\beta_j$ is the regression coefficient (with $c_j$ being an intercept term), and is analogous to the $f$ value from the crossing fibres model; i.e., the fraction of signal contributed by $\bar{v}_j$. We refer to these coefficients as *tract-specific anisotropy* (TSA).

**Statistics and reproducibility**. The TSA values obtained in the preceding step were warped back into standard space and smoothed with a Gaussian kernel with a full-width at half-maximum (FWHM) of 1.5 mm. Subsequently, for each voxel in a tract, the following linear regression model was tested, for the first-order effects of *Age* and *Sex*, and their interaction:

$$\text{TSA} = \beta_0 + \beta_1 \cdot \text{Age} + \beta_2 \cdot \text{Sex} + \beta_3 \cdot \text{Age} \times \text{Sex} + \varepsilon \quad (4)$$

For each contrast, we next wanted to summarise the resulting t-statistics at each discrete distance along the trajectory of the tract. To do this, following an approach similar to TBSS[8], we computed weighted t-statistics for each voxel $j$ as $t'(j) = t(j) \cdot \mathbf{P}_{ab-\text{tract}}(j)^\lambda$, where $\lambda$ determines the rate of decay (here, we set $\lambda = 1.0$). At each discrete distance $d$ along the tract, we assigned (unweighted) $t_d$ as the t-statistic corresponding to the maximal $t'_d$.

Because anatomical properties along a tract can be assumed to form a continuous random field (i.e., neighbouring vertices have a spatial dependence), we analysed the resulting distance-wise summary statistics as a one-dimensional random field, using the `rft1d` Python library (http://www.spm1d.org/rft1d/;[82]). Firstly, the spatial smoothness of model residuals was estimated for each tract as a FWHM value, and averaged across tracts. For each tract, this mean FWHM was used to compute $t^*$, the critical t-value at $\alpha = 0.05$ for a Gaussian field, using an inverse survival function. Secondly, cluster-wise inference was performed to identify significant clusters along the tract, with a minimum cluster size of 3. Thirdly, p-values for all clusters, across all tracts, were corrected for family-wise error using a false discovery rate (FDR) threshold of 0.05. FDR was performed using the `statsmodels` Python library, with a two-stage non-negative FDR method (`fdr_tsbky`). Finally, to estimate the spatial extent of effects for a given tract, significant t-values were summed over that tract, for positive and negative t-values separately.

**Reporting summary**. Further information on research design is available in the Nature Research Reporting Summary linked to this article.

## Data availability
The Enhanced NKI-Rockland dataset[73] used in this study is freely available at www.nitrc.org/projects/fcon_1000/. Derived participant- and sample-wise data, along with parameter files specifying the processing and analysis steps used to generate them, are freely available from the University of Nottingham Research Data Management Repository at https://doi.org/10.17639/nott.7102. Source data for the DMN scatterplots in Fig. 6 and Supplementary Fig. 3, and t-value plots in Supplementary Fig. 1, are provided as comma-separated values in Supplementary Data 1. Source data for the WWN scatterplots in Fig. 6 and Supplementary Figs. 4 and 5, and t-value plots in Supplementary Fig. 2, are provided as comma-separated values in Supplementary Data 2.

## Code availability
All code for this procedure was written in Python 3 (Anaconda build), making system calls of native FSL packages (available at https://fsl.fmrib.ox.ac.uk/), as indicated. The following third-party Python libraries were also used: `nibabel`, `nilearn`, and `statsmodels`. Source code for all procedures used in this study is freely available as `dwi-tracts` (v.1.0.2)[83]. This includes: scripts to run FSL preprocessing and DWI analyses, using standard SGE- or SLURM-style parallel environments; a module to estimate core trajectories and TSA values; a module to fit general linear models to TSA values; and a module to plot the results of these. The output of these modules (polylines, surface meshes, graphs) can be visualised using ModelGUI software, an open source Java library available at https://github.com/neurocoglab/mgui-core.

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

## Acknowledgements

We thank Saad Jbabdi for extremely helpful feedback, in particular regarding the TSA approach, as well as Marije ter Wal and Christopher Madan for their invaluable advice on various parts of the manuscript. 2D and 3D renderings for this publication were generated using ModelGUI, a free and open source software package (http://www.modelgui.org/). Parallel processing at the University of Nottingham was provided through its Augusta HPC service and the Beacon of Excellence for Precision Imaging, which provide access to high performance computing resources for the University's neuroimaging research community. This study was supported by the National Institute of Mental Health (R01-MH074457), the Helmholtz Portfolio Theme "Supercomputing and Modeling for the Human Brain" and the European Union's Horizon 2020 Research and Innovation Programme under Grant Agreement No. 945539 (HBP SGA3).

## Author contributions

A.T.R. wrote and tested all Python code, produced estimates and analysis results using the NKI dataset, and composed the primary text and figures. J.A.C. was involved in early data exploration, code development, and visualisation of individual tracts. F.H. was involved in idea generation, provided extensive input into the methodological approach, and was involved in the development of the initial tract estimation algorithms. S.B.E. originated the study concept, wrote the original (Matlab) code for tract estimation, and

provided continuous feedback and guidance on all aspects of the study, including manuscript composition and figure design.

## Competing interests

The authors declare no competing interests.
