## [Transparent Peer Review File · Communications Biology]

Reviewers' Comments:

Reviewer #1:

Remarks to the Author:

Large databases of neuroimaging require automated processing. Diffusion-weighted imaging tractography that allows assessing white matter integrity in the living human brain is no different. Early work provided solid methods to do so 'tract-based spatial statistics' TBSS that allows for the systematic and statistical comparison of the skeleton of white matter between subjects. However, as stressed by the authors' interpretation of the findings from TBSS are limited as the origin and the termination of the white matter regions deemed significant are not available with that methods. Further, albeit TBSS updates can take into account fibre crossing, they are still limited and poorly used by researchers.

Therefore in this manuscript, Reid et al. propose an alternative approach "Tract-specific statistics based on diffusion-weighted probabilistic tractography". Data and code are made openly available for others to use.

I think this is a great initiative, and I fully support this manuscript.

Some important points.

As it read, the authors seem to introduce this work as the first attempt to measure tract specific anisotropy. However, this is not the case as other authors have been developing measures and software that are fibre specific. 2 important references regarding that matter:

Raffelt D, Tournier JD, Rose S, Ridgway GR, Henderson R, Crozier S, Salvado O, Connelly A. Apparent Fibre Density: a novel measure for the analysis of diffusion-weighted magnetic resonance images. *Neuroimage*. 2012 Feb 15;59(4):3976-94. doi: 10.1016/j.neuroimage.2011.10.045. Epub 2011 Oct 20. PMID: 22036682.

Dell'Acqua F, Simmons A, Williams SC, Catani M. Can spherical deconvolution provide more information than fiber orientations? Hindrance modulated orientational anisotropy, a true-tract specific index to characterise white matter diffusion. *Hum Brain Mapp*. 2013 Oct;34(10):2464-83. doi: 10.1002/hbm.22080. Epub 2012 Apr 5. PMID: 22488973; PMCID: PMC6870506.

Since age differences was targeted as a proof of concept in the present paper, the manuscript should include references to the pioneers in the field of age-related diffusion weighted imaging differences.

Lebel C, Treit S, Beaulieu C. A review of diffusion MRI of typical white matter development from early childhood to young adulthood. *NMR Biomed*. 2019 Apr;32(4):e3778. doi: 10.1002/nbm.3778. Epub 2017 Sep 8. PMID: 28886240.

Hsu JL, Leemans A, Bai CH, Lee CH, Tsai YF, Chiu HC, Chen WH (2008) Gender differences and age-related white matter changes of the human brain: a diffusion tensor imaging study. *Neuroim- age* 39(2):566-577

Hasan KM, Kamali A, Iftikhar A, Kramer LA, Papanicolaou AC, Fletcher JM, Ewing-Cobbs L (2009b) Diffusion tensor tractography quantification of the human corpus callosum fiber pathways across the lifespan. *Brain Res* 1249:91-100

Lebel C, Beaulieu C (2011) Longitudinal development of human brain wiring continues from childhood into adulthood. *J Neurosci* 31(30):10937-10947

Bastin ME, Piatkowski JP, Storkey AJ, Brown LJ, MacLullich AM, Clayden JD (2008) Tract shape modelling provides evidence of topological change in corpus callosum genu during normal ageing.

Neuroimage 43(1):20–28

Michielse S, Coupland N, Camicioli R, Carter R, Seres P, Sabino J, Malykhin N (2010) Selective effects of aging on brain white matter microstructure: a diffusion tensor imaging tractography study. *Neuroimage* 52(4):1190–1201

The application of a fibre specific anisotropy measurement to ageing also demonstrated higher sensitivity and specificity than traditional fractional anisotropy measurements in the following reference.

Rojkova K, Volle E, Urbanski M, Humbert F, Dell'Acqua F, Thiebaut de Schotten M. Atlasing the frontal lobe connections and their variability due to age and education: a spherical deconvolution tractography study. *Brain Struct Funct.* 2016 Apr;221(3):1751-66. doi: 10.1007/s00429-015-1001-3. Epub 2015 Feb 15. PMID: 25682261.

The methods regarding the tractography are missing a few details.

How many fibres populations were modelled per voxels? 2 or 3? This is important to specify because it has an impact on the solving of the crossing.

"Distance correction was applied", it is essential to specify that distance correction is a multiplication of the tract probability by the distance. It does not really solve the issue of distance and structural connectivity.

Some limitations should be acknowledged:

- It is doubtful that this new approach can be applied to the whole brain due to the penalisation for multiple comparisons that increase exponentially with the increase of the number of region of interest.
- The measurement can be biased (as many other methods) by aligning to the stereotaxic space that, when not perfect, can lead to increased variability in the findings.
- Some indication on the computation time, which is believed to be rather long, should be mentioned.

Other than that, excellent work.

Michel Thiebaut de Schotten

Reviewer #2:

Remarks to the Author:

This work presents a method to analyze the integrity of white matter tracts in vivo from diffusion MRI images. This method is based on post-processing probabilistic tractography-issued tracts and then aggregate the tracts across different subjects to obtain a weighed mask that then is used to define a tract-specific anisotropy estimations. This technique is then used to determine sex and age effects on tracts connecting regions belonging to the default mode network (DMN) and the what-where network (WWN). The applications are well designed and presented. My review will mostly be centered on the technical aspects of diffusion MRI.

My first point is with respect to the innovative aspect of the method. Specifically, aggregating streamlines across subjects to build a population-specific weighted tract mask has been proposed more than 10 years ago by Hua et al (*Neuroimage* 2007), even to compute the "tract-specific anisotropy" proposed in this article. This technique has been then used by many different studies (see e.g., Jolles et al 2014, Subramaniam et al 2018) and proposed in several different variants. A succinct review of some of these proposals and studies is presented in Calamante (*Magn Reson Mater Phy* 30, 317–335, 2017). It should be noted that as the authors are basing their analysis on streamlines, the procedure is roughly similar for deterministic or probabilistic tractography. Furthermore, the method proposed to select the most probable tract, is admittedly heuristic and composed of a multi-step protocol where different parameters are hand-adjusted, which opens questions on the

reproducibility of the analysis. In all the method to obtain the most representative tract and to aggregate the tract-specific anisotropy estimation is, without acknowledging it, a re-iteration of methods that have been proposed at-length in the past 10 years in the diffusion community.

My second point, and a less important one, is with respect to the introduction. There are statements in the introduction which are plainly wrong an example of this being: "for two identical fibres oriented along the anteroposterior (AP) axis, FA would be inversely proportional to the number of fibres crossing each along the (perpendicular) mediolateral (ML) axis". FA is a complex and diffuse aggregate measurement depending on several physiological factors, such as density of axons in a specific voxel, proportion of extra-cellular water, among other phenomena (see Wang et al Brain 2011 for an initial evaluation of this and Novikov et al 2018, for a review of this phenomenon).

In all, the results proposed as an example can be of value to the community. However, this article is presented as a novel method to compute tract-specific measurements which, at least in the current presentation does not appear to be different from a long tradition of proposed methods, and the presentation of diffusion MRI specifics in the introduction fails at portraying the authors as experts in the field.

Response to Reviewers

General

Both reviewers have raised excellent points about previous advances in DWI tractography-based approaches, and particularly those that target tract specificity. We agree that our initial introduction to the current study lacked a thorough discussion of the preceding body of work, and have substantially amended both the Introduction and Discussion sections to address this shortcoming. Additionally, the term “tract” should also be better defined as the set of axons projecting (possibly bidirectionally) between two grey matter regions, which is the main point of differentiation between the current approach and others highlighted by both reviewers. However, we do not feel that the present study is merely an iteration of existing methods, at least not based on any studies we are aware of. A comparison of the specific studies highlighted by the reviewers is given below, and where appropriate has also been incorporated into our Introduction and Discussion section. We are grateful to the reviewers for suggestions that have, in our opinion, improved these sections.

Reviewer 1

1. As it read, the authors seem to introduce this work as the first attempt to measure tract specific anisotropy. However, this is not the case as other authors have been developing measures and software that are fibre specific.

This is certainly true, and we thank reviewer for making us aware of these two studies in particular. We did not intend to represent the current study as the first such attempt at fibre-specific approaches, and agree with both reviewers that our introduction should be modified to properly summarise the existing literature on this topic. The relevant Introduction section (page 2) now includes this discussion as an appended paragraph:

Additional fibre-specific DWI approaches have also been proposed, including q-space and q-ball imaging, spherical deconvolution, and CHARMED (Tuch, 2004; Tournier et al., 2004; Assaf and Basser, 2005). In particular, spherical deconvolution uses a spherical harmonic decomposition to estimate an orientation distribution function (ODF) from the observed diffusion signal (Tournier et al., 2004). This approach allows both the diffusion model and the number of distinct fibre populations within a voxel to be estimated from the observed data, and is the basis for voxel-wise estimation of apparent fibre density (AFD) for these distinct populations (Raffelt et al., 2012). Differences in oriented clusters of AFD (also referred to as fixels; Raffelt et al., 2015) have been shown for patients with motor

neurone disease (Raffelt et al., 2012) and Alzheimer's disease (Mito et al., 2018). A related spherical deconvolution-based approach, called hindrance modulated orientational anisotropy (HMOA), uses the amplitude of specific lobes of the ODF as an estimate of white matter integrity for a specific fibre population (Dell'Acqua et al., 2012). HMOA of the postcommissural fornix has been shown to predict verbal memory performance in a healthy aging cohort (Christiansen et al., 2016).

Additionally, we have modified the Discussion in other to better acknowledge how these existing approaches may complement our the proposed methodology (page 12; changes underlined):

It is possible that the increased specificity of the current approach permits a more fine-grained spatial and angular dissection of effects than does TBSS with FA, which uses a more coarse-grain white matter skeleton, and is not orientation-specific. If so, then the positive effects observed here may reflect a real age-related increase in white matter integrity for specific tracts. This possibility is supported by reports of fairly widespread increases in fMRI-based functional covariance with aging (Tomasi and Volkow, 2011), which have been proposed to reflect compensatory changes in response to degeneration or dysfunction of other brain regions. Given the conflicting evidence, however – particularly with the HMOA evidence from Rojkova et al. (2015), which is fibre-specific – these effects should be interpreted with caution. It will be important in future TSA studies to increase the number of ROIs, or query specific crossing tracts, in order to obtain a more complete picture of age-related effects across white matter. One promising avenue could be based on the so-called "tract-specific fractional anisotropy" approach (Mishra et al., 2014), in which uses the free water fraction to estimate an adjusted FA value in crossing-fibre regions. Alternatively, an integration of the current approach with existing fibre-specific spherical deconvolution-based methods, such as AFD (Raffelt et al., 2012) or HMOA (Dell'Acqua et al., 2012), could potentially be used to disambiguate the interpretation of TSA values in areas of dense crossing fibres.

2. Since age differences was targeted as a proof of concept in the present paper, the manuscript should include references to the pioneers in the field of age-related diffusion weighted imaging differences.

The application of a fibre specific anisotropy measurement to ageing also demonstrated higher sensitivity and specificity than traditional fractional anisotropy measurements in the following reference.

We agree that the discussion of age effects should be expanded. We have included a more expansive discussion of the relevant studies, only omitting those suggestions

that were aimed at childhood and adolescence, since these don't overlap with the age range of our cohort. The relevant Discussion section now reads (page 11; changes underlined):

For the WWN, age-related decreases in TSA occurred mainly in the body, but not the splenium, of the corpus callosum. This pattern is in agreement with several DWI-based studies of age-related connectivity changes. Burzynska et al.(2010) used TBSS to show an age-related reduction in FA (and increase in radial and mean diffusivity) in the genu and body of the corpus callosum, but not the splenium. Using a DTI approach, Bennett and Madden (2014) found decreased FA in older versus younger participants for both the genu and splenium, but with a more pronounced effect in the former. The same authors report an anterior-to-posterior gradient in age-related FA changes, with these being more pronounced in frontal white matter, consistent with the pattern found in the current study (Bennett et al., 2009). An even more pronounced pattern was reported by Michielse et al. (2010), who found an age-related decrease of FA in the genu, no relationship in the body, and a late (age 70-85) increase in the splenium. Bastin et al.(2008), also using tract shape modelling, found a significant decrease in FA for the genu, but not the splenium, in an elderly cohort (age 65 to 87), while in a cohort ranging from 30 to 80 years, Hsu et al. (2008) reported a similar age-related decrease in FA (and increase in MD) for the anterior, but not posterior corpus callosum. Interestingly, evidence from Hasan et al. (2009) suggests FA in the corpus callosum changes in a quadratic manner across the lifespan: increasing between age 7 and 20 years and decreasing between 20 and 60, which is in line with the present findings. Taken together, these findings suggest that, in adults, age-related changes in WM integrity of the corpus callosum may be more prominent anteriorly, and reduced or even reversed posteriorly.

Positive age associations were a more surprising finding, as numerous articles report negative age/FA associations (e.g., Kodiweera et al., 2016) and postmortem evidence of white matter loss and decrease in the proportion of small myelinated fibres with age (Tang et al., 1997; Aboitiz et al., 1997). Both positive and negative age/FA associations have been reported in at least one previous brain-wide TBSS study (Kochunov et al., 2007), however. For DMN in particular, we found that most positive associations occurred in regions proximal to ROIs, where the potential confound of crossing fibres is likely more pronounced (Jeurissen et al., 2012). The positive relationships in the WWN were especially prominent, and suggest a (paradoxical) increase in white matter integrity in these tracts. The majority of positive relationships in WWN were found in the middle of the superior longitudinal fasciculus (SLF). One TBSS study focusing on the SLF in a healthy cohort found no effect of age on FA (Madhavan et al., 2014), while another whole-brain study found SLF among tracts with negative age effects (Marstaller et al., 2015).

Similarly, Rojkova et al. (2015), using FA and the fibre-specific HMOA approach, found that both metrics decreased with age in the SLF bilaterally. On the other hand, increased FA in Alzheimer's disease patients has also been reported in SLF (Douaud et al., 2011). The authors of this study suggest that a relative sparing of crossing motor fibres may account for this effect, but this is inconsistent with our observed increase in TSA; on the contrary, our findings might be explained by a relative *decrease* in WM integrity in these crossing fibres.

3. The methods regarding the tractography are missing a few details.

How many fibres populations were modelled per voxels? 2 or 3? This is important to specify because it has an impact on the solving of the crossing.

The default of N=3 populations was used in this study; this has now been added to our Preprocessing section (page 14).

4. "Distance correction was applied", it is essential to specify that distance correction is a multiplication of the tract probability by the distance. It does not really solve the issue of distance and structural connectivity.

We totally agree with this; the parameter was used in order to ensure that the resulting distributions were less highly skewed towards voxels near to the seed ROI. The relevant section has been updated to emphasize that this correction is not meant to actually solve the issue of distance bias (page 14; changes underlined):

Probabilistic tractography was performed in MNI-152 space, using the `probtrackx` command (Behrens et al., 2007). For both networks, we generated 50,000 streamlines per seed voxel, with the other seed regions as target masks. We applied a step length of 0.5 mm, a curvature threshold of 0.2, a minimal path distance of 5 mm, and a fibre threshold of 0.01. A distance correction was also applied. Notably, this correction merely multiplies the streamline count at each voxel by its expected distance from the seed voxel. This is not an adequate correction for distance bias (which is unlikely to be linear), but does help reduce this bias in the middle of trajectories, for the purpose of tract determination (see next section). For each voxel, `probtrackx` recorded the number of streamlines that encountered that voxel, producing a separate count for streamlines terminating in each target ROI. Additionally, the voxel-wise mean orientation of streamlines between each seed/target pair was computed.

5. It is doubtful that this new approach can be applied to the whole brain due to the penalisation for multiple comparisons that increase exponentially with the increase of the number of region of interest.

While we hope that this method might eventually be refined in a way that accommodates whole brain analyses, in its present form we agree that this will have a high cost in terms of family-wise error. Notably, however, the approach (and number of tests) should scale quadratically with the number of ROIs, and an approach such as one-to-all, rather than all-to-all, could be feasibly applied. We also take hope from the use of FDR control in GWAS studies, where millions of tests are performed. A more thorough discussion of this complexity issue is now provided as a new paragraph in the Discussion (page 13):

The relatively small networks used here (9 ROIs for DMN and 10 ROIs for WWN) required a total processing time of over 200 hours per participant, on CPU processors (see Supplementary Table 3). Notably, processing time will scale quadratically with the number of ROIs ($O(n^2)$), indicating that it may be infeasible to apply the TSA approach to the full set of possible ROIs (of comparable size) in the brain. On the other hand, the volume of grey matter in the brain is finite, and the number of ROIs comprising a whole-brain network depends critically on their *granularity*. Additionally, the bulk of processing time for this approach is attributable to the preprocessing steps (bedpostx and probtrackx), which were run using CPUs. GPU versions of these functions have recently been introduced, which can reduce processing time by a factor of 200 (Hernandez-Fernandez et al., 2019). This would reduce the required processing time for the present study to 10 hours per participant. A related limitation is that the number of multiple hypothesis tests (and associated family-wise error) also increases by $O(n^2)$. However, the false discovery rate (FDR) approach we use to control family-wise error should be robust even to the high number of tests expected with a whole-brain TSA analysis (and is commonly used in genomic studies; see Korthauer et al. 2019). Ultimately, whether whole-brain TSA analysis is feasible in practice remains to be demonstrated.

6. The measurement can be biased (as many other methods) by aligning to the stereotaxic space that, when not perfect, can lead to increased variability in the findings.

This is indeed an important caveat, and is especially worrisome when applied to neurodegeneration, neurodevelopment, or lesions, where the structure of the brain and the associated normalization can be expected to change substantially. This may be especially relevant near ventricles, as is seen in Figure 4b. We have modified our discussion of this potential limitation (which is also an issue for many neuroimaging methods) in the Discussion (page 12; changes underlined):

The NKI Rockland dataset was chosen due to its large size, age range, and the use of a single MRI scanner and protocol. To ensure the cohort was as representative

of the general population as possible, and to enable the analysis of age over the lifespan, we chose to use close to the full age range (18-80), and to exclude participants with clinical diagnoses. As with most population templates, however, the choice of cohort is an important consideration when interpreting a derived result. The human brain is known to show systematic anatomical grey matter changes across the lifespan (Sowell et al., 2003; Giorgio et al., 2010), and this will almost certainly bias normalization in a way that may account for a portion of the TSA effects reported here. Indeed, variability of findings due to the choice of T1w templates has been shown for voxel-based morphometry (Shen et al., 2007). It will be important in future studies to assess the influence of this bias, use cohorts that are more targeted to a particular phenomenon under investigation, generate population-specific T1w templates for normalization (see, e.g., Whitwell et al., 2007), and compare the predictions of TSA to in vivo or post mortem analyses of white matter (e.g., as in Reveley et al., 2015).

7. Some indication on the computation time, which is believed to be rather long, should be mentioned.

This information has been added as a new Supplementary Table 3 (page 24):

Supplementary Table 3. Typical processing times for a single subject on a CPU core (note that BedpostX is only run once).

Network	Step	Processing time (hours)
DMN	BedpostX	40.3
	ProbtrackX	75.0
	Tract determination	3.0
	TSA computation	0.2
	GLM analysis	1.1
	Total	119.6
WWN	BedpostX	40.3
	ProbtrackX	83.4
	Tract determination	3.2
	TSA computation	0.3
	GLM analysis	1.3
	Total	128.5
Total both		207.8

A discussion of this has also added to the Discussion, highlighting the drastic performance improvements (approximately 200x) supplied by GPU implementations of the FDT tools (see Hernandez-Fernandez et al., 2019) (page 13; see above).

Reviewer 2

1. My first point is with respect to the innovative aspect of the method. Specifically, aggregating streamlines across subjects to build a population-specific weighted tract mask has been proposed more than 10 years ago by Hua et al (Neuroimage 2007), even to compute the “tract-specific anisotropy” proposed in this article.

We thank the reviewer for highlighting this article, of which we were not aware. It does indeed apply an approach for averaging streamlines across a cohort and characterize FA along their trajectory. However, it is our opinion that our current approach diverges in a number of fundamental ways from that described by Hua et al. (2007):

1. The authors use deterministic tractography (FACT, Mori et al., 1999) rather than probabilistic. The advantage of the latter to our approach is that it is much more likely to produce a diffuse rather than restricted streamline probability distribution, which is especially important for detecting bidirectional overlap (see Petersen et al., J Neurosurg 2016; Schlaier et al., Eur J Neurosci, 2017).
2. They define 11 large *white matter* ROIs, while our method accepts arbitrary, relatively small *grey matter* ROIs. Their method is therefore not tract-specific in the same sense as ours, as it cannot assign an anisotropy metric to individual ROI pairs. The sense of the term “tract” is critical in this respect, and we have updated our text to specify this (see below).
3. Relatedly, their method does not consider the overlap in *bidirectional* tractography, i.e., from two seed ROIs. As stated in (1), probabilistic tractography is advantageous for this as it more thoroughly samples the diffusion space.
4. The authors derive FA measures from the DTI model, but *not* by regressing diffusion parameters against the average tract orientation (our equation 3), which is a key aspect of our approach that makes it more tract-specific.
5. Equations (1) and (2) in that article yield a single weighted average for each full tract, rather than a geometrically-informed mapping onto discrete distances along the tract. Our measure is thus *spatially resolved*, and can be used to localize where along a specific tract a particular effect is occurring (Figure 5).

We agree that this study is an important precursor to ours, and have added a discussion of this article, along with these points of divergence, in our Discussion section (page 10-11, changes underlined):

Previous studies have investigated DWI metrics in a tract-specific manner. Notably, Hua et al. (2008) produced probabilistic maps of 11 gross WM tracts by seeding in WM voxels and averaging deterministic tractography streamlines across participants, generating tract-specific metrics by averaging FA across all voxels in a tract. The well-known TBSS approach (Smith et al., 2007), in which statistics are

projected onto a pre-established population-based white matter “skeleton”, similarly allows DWI metrics to be mapped to specific WM tracts. Both of these approaches are similar to the current method in that they utilize population averaged images (streamline counts or FA values) in order to generate probabilistic maps of WM tract geometry (see Calamante, 2017, for a review of such approaches). These maps serve a similar function to population-based anatomical grey matter templates, such as the linear and nonlinear ICBM-152 templates (Fonov et al., 2009; Mazziotta et al., 1995). There are two major advantages of the present method over the Hua et al. (2008) method and TBSS: (1) it allows a population-based tract estimate to be derived specifically for the white matter tract connecting any two arbitrarily-defined GM ROIs, if that tract is likely to exist; and (2) it allows participant- and tract-specific anisotropy to be estimated, based on the orientation of streamlines defining that tract in each voxel along its trajectory.

2. This technique has been then used by many different studies (see e.g., Jolles et al 2014, Subramaniam et al 2018) and proposed in several different variants.

We were unfortunately not able to identify Jolles et al. (2014), but we have evaluated Subramaniam et al. (2018). In this article, the authors apply the TBSS approach to their analysis of training-related white matter changes in schizophrenia. TBSS generates an FA “skeleton” by thresholding an average FA map, and then maps individual FA values onto this map to facilitate statistical analysis (Smith et al., 2006; 2007). Notably, we have discussed TBSS in the manuscript as the most closely related method to the one we propose, and highlighted in detail how our method extends it (see also the comments of Reviewer 1 on this relationship). Specifically, our method:

1. produces spatial trajectories specific to pairs of arbitrarily defined GM ROIs
2. produces tract-specific anisotropy based on the average streamline orientation for each voxel in a specific trajectory; i.e., specific to that ROI pair
3. maps statistics onto this trajectory in a manner similar (but not identical) to TBSS

3. A succinct review of some of these proposals and studies is presented in Calamante (Magn Reson Mater Phy 30, 317–335, 2017).

This is a comprehensive review of approaches for averaging streamlines, typically generated from brain-wide WM. We appreciate being made aware of the extensive literature on this topic, which is relevant for the first step of our methodology. However, the majority of studies described focus on seeding in WM voxels, and as far as we can tell, none describe a method for integrating bidirectional averaged streamlines generated from ROI pairs, estimating core trajectories from this integrated probability map, or generating an voxel-wise anisotropy estimate specific to the average

orientations of streamlines used to produce that map. Indeed, as far as we can tell, this is a novel method and diverges substantially from existing literature.

To better acknowledge the existing work done to characterize and map average streamlines, we have added a reference to this review article to our Discussion section (page 10, see previous quote).

4. It should be noted that as the authors are basing their analysis on streamlines, the procedure is the roughly similar for deterministic or probabilistic tractography.

The current method could certainly be applied to deterministic tractography as well as probabilistic. We expect that the outcomes will differ in important ways for the two approaches. The use of probabilistic tractography was chosen largely because it can reasonably be expected to more thoroughly sample the space of possible streamlines, which is important especially for identifying bidirectional overlap. Studies comparing the two methods substantiate this (e.g., Zolal et al., J Neurosurg, 2016: <https://doi.org/10.3171/2016.8.JNS16363>; Schlaier et al., Eur J Neurosci., 2017: <https://doi.org/10.1111/ejn.13575>).

5. Furthermore, the method proposed to select the most probable tract, is admittedly heuristic and composed of a multi-step protocol where different parameters are hand-adjusted, which opens questions on the reproducibility of the analysis.

There are indeed a number of parameters that define each step of this method. The most relevant are briefly discussed below:

1. A threshold is applied to the bidirectional average map in order to identify a single trajectory from potentially many. This threshold was “hand-adjusted” in the sense that we determined a threshold that, for most tracts, achieved this objective. All tracts were produced using the same threshold, however, and applying thresholds to neuroimaging maps is a common step in many approaches (including TBSS).
2. Angle and length constraints are applied to the core tract polyline generation process. These are used to prevent the generation of unrealistic polylines, especially in regions close to ROIs where probabilities are more diffuse. Again, these constraints were determined as those that best achieved this objective, and were the same for all tracts. Such constraints are common for streamline generation (including ProbtrackX).
3. Parameters are used to define the Gaussian field that is used to generate the final tract estimation; i.e, by multiplying with the original probability map. These were chosen to constrain the spatial extent of the tract in a smooth way.

4. TSA statistics are mapped to the core trajectory by weighing based on the tract map (see Line 236). A “lambda” parameter is provided that can influence the spatial distribution of this weighting, although we use lambda=1 and this parameterization is completely optional.

Importantly, each of these parameters is simply a way to fine-tune the method to attain the heuristical objective: “we want a spatially constrained distribution that represents the most probable tract, based on the diffusion evidence, between two GM ROIs”. It is unlikely, in our opinion, that changes to these parameters will significantly alter the observed results, other than to produce failures for a higher fraction of ROI pairs, or alter the spatial extent (or smoothness) of the resulting statistics. We also note that many existing and well-established neuroimaging methods use parameters in this way (e.g., TBSS, ProbtrackX, BET, Freesurfer, etc.).

In our opinion, using the default parameters specified in the manuscript will result in outcomes that are reproducible and comparable. We only recommend altering these parameters for a specific purpose; e.g., to broaden or shrink the spatial extent of the resulting TSA map.

This is indeed an important consideration, and we have added a new paragraph to our Discussion section (page 12-13):

The estimation of tract trajectories and TSA values involves a heuristic approach, with numerous parameters involved at each step. This raises the potential for parameter adjustments to vary the results in ways that bias the resulting distributions and statistics. In general, however, these parameters were chosen in order to spatially constrain these estimates in reasonable ways; for example: to optimise a threshold such that a single trajectory is chosen from several alternatives, to constrain core polylines to realistic geometries, or to determine the spatial extent at which TSA values are used to compute distance-wise statistics. Of these parameters, it is only the latter where the researcher is able to exercise discretion, i.e., over the degree of spatial certainty used to map statistics at a given distance to the “core” tract trajectory (specified by the σ_r , and λ parameters). As a general policy, we recommend using the default parameter values for tract and TSA estimation, in the absence of a principled reason to adjust them

6. In all the method to obtain the most representative tract and to aggregate the tract-specific anisotropy estimation is, without acknowledging it, a re-iteration of methods that have been proposed at-length in the past 10 years in the diffusion community.

As argued above for the specific examples given, and based on our knowledge of the current literature, we respectfully disagree that it is a “re-iteration” of any existing method.

7. My second point, and a less important one, is with respect to the introduction. There are statements in the introduction which are plainly wrong an example of this being: “for two identical fibres oriented along the anteroposterior (AP) axis, FA would be inversely proportional to the number of fibres crossing each along the (perpendicular) mediolateral (ML) axis”. FA is a complex and diffuse aggregate measurement depending on several physiological factors, such as density of axons in a specific voxel, proportion of extra-cellular water, among other phenomena (see Wang et al Brain 2011 for an initial evaluation of this and Novikov et al 2018, for a review of this phenomenon).

While we accept that the anatomical properties generating an FA signal are more complex than simply oriented axonal compartments, the sentence above is intended to present a simple hypothetical situation that challenges the use of streamline counts to infer connectivity density. Two “identical fibres” will generate different streamline distributions if one is crossed by a perpendicular fibre and the other is not. Moreover, the relationship will be inversely proportional because the crossing fibre will reduce FA proportional to the fraction of diffusion occurring along its axis. This follows directly from the formula for FA (Basser & Pierpaoli, 1996):

$$FA = \sqrt{\frac{3}{2} \left(\frac{(\lambda_1 - \hat{\lambda})^2 + (\lambda_2 - \hat{\lambda})^2 + (\lambda_3 - \hat{\lambda})^2}{\lambda_1^2 + \lambda_2^2 + \lambda_3^2} \right)}$$

If diffusivity in the perpendicular direction (λ_2) is varied from 0 to 1, while holding λ_1 constant, the relationship with FA is indeed inversely proportional:

The issue is perhaps the wording of the sentence, which we have modified to reflect the complexities subsumed by the word “identical” (page 2, changes underlined):

It is often desirable to relate voxel-wise DWI-based metrics such as FA to other phenotypical observations, such as behavioural or cognitive measures, or clinical status. Voxel-wise analyses can be highly confounded by the individual geometry of white matter tracts, and one way to address this issue is tract-based spatial statistics (TBSS), in which FA measures are projected onto a population-based FA “skeleton” with a high probability of being white matter in all participants (Smith et al., 2006,2007). The presence of crossing fibres, however, also has implications for the interpretation of FA (Jbabdi et al., 2010). As a hypothetical example: for two otherwise anatomically identical white matter fibres, the introduction of perpendicularly oriented axons to one will reduce its FA proportionally. Interpreting FA in terms of the underlying microstructure of white matter in a voxel is thus inherently ambiguous. This ambiguity can be improved if crossing fibres are explicitly modelled, for example using the Bayesian approach described above. Such a crossing fibre model has been proposed as an extension to the TBSS approach (Jbabdi et al., 2010).

We are happy to respond to any other perceived issues with statements made in the revised Introduction.

REVIEWERS' COMMENTS:

Reviewer #1 (Remarks to the Author):

The authors appropriately revised the manuscript.

Just a minor point. The mention about the work of Rojkova is not correct. I would rather suggest the following:

Similarly, Rojkova et al. (2015), using FA and the fibre- specific HMOA approach demonstrated that fiber specific was more sensitive than FA to age related changes particularly when involving crossing fibres.